# First Gut Content Analysis of 4th Instar Midge Larvae (Diptera: Chronomidae) In Large-Scale Weirs Using a DNA Meta-Barcoding Approach

**DOI:** 10.3390/ijerph17082856

**Published:** 2020-04-21

**Authors:** Hyunbin Jo, Bohyung Choi, Kiyun Park, Won-Seok Kim, Ihn-Sil Kwak

**Affiliations:** 1Fisheries Science Institute, Chonnam National University, Yeosu 59626, Korea; prozeva@jnu.ac.kr (H.J.); chboh1982@chonnam.ac.kr (B.C.); ecoblue@hotmail.com (K.P.); 2Division of Fisheries and Ocean Science, Chonnam National University, Yeosu 59626, Korea; csktjr123@gmail.com

**Keywords:** gut content, Chironomidae larvae, large-scale weirs, DNA meta-barcoding, feeding behavior

## Abstract

Chironomidae larvae play an important role in the food chain of river ecosystems in Korea, where it is dominant. However, detailed information on the diet of Chironomidae larvae are still lacking. The purpose of this study was to identify the gut contents of 4th instar larvae of a Chironomidae inhabiting four large-scale weirs (Sejong Weir, Juksan Weir, Gangjeong-Goryeong Weir, and Dalseong Weir) using a DNA meta-barcoding approach. We found that dominant Operational Taxonomic Unit (OUT) was assigned to *Paractinolaimus* sp. (Nematoda), and the sub-dominant OTU was assigned to *Dicrotendipes fumidus* (Chironomidae). The most common OTUs among the individuals included phytoplankton, such as *Tetrahymena* sp., *D. armatus*, *Pseudopediastrum* sp., *Tetradesmus dimorphus*, *Biddulphia tridens*, and *Desmodesmus* spp. We calculated the selectivity index (*E*’) and provided scientific evidence that Chironomidae larvae have a significant preference (*E*’ > 0.5) for *Desmodesmus armatus*, *E. minima*, and *T. dimorphus*, while it does not show preference for other species found in its gut. Differences in physico-chemical factors, such as water quality, nutrients, Chl-*a*, and carbon concentrations, resulting from anthropogenic impacts (i.e., construction of large-scale weirs) as well as the particle size of prey organisms (small-sized single cell) and effects of chemicals (chemokinesis) could affect the feeding behavior of Chironomidae larvae.

## 1. Introduction

Chironomidae is a large group of invertebrates, with a reported diversity of 8000–20,000 species, and its members are distributed worldwide [1]. Chironomidae adults inhabit areas near the riparian zone of rivers or lakes, but the larvae are aquatic organisms that are distributed in diverse aquatic habitat patches [2]. The 1st instar larvae starts its lifecycle by settling on the water surface of aquatic ecosystems after hatching from its egg [1]. The 3rd or 4th instar in the bottom substrates has a formed cage [1] and starts to filter phytoplankton and predate on organisms, such as zooplankton and other small organisms [3,4,5]. The life cycle of a Chironomidae larvae is sensitive to anthropogenic impacts, such as changes in habitat traits and water quality, and it is also important to fish and birds as a food source [6,7]. Therefore, identifying the food of the 3rd or 4th instar larvae of a Chironomidae species is critical for understanding the role of Chironomidae in aquatic ecosystems.

Recently, 16 large-scale weirs were built along the main channels of the four largest rivers (the Han, Nakdong, Geum, and Yeongsan) in South Korea to stimulate development of water resources for recreation and other purposes [8,9,10]. The construction of the weirs, which involved the dredging of the riverbed, channelization, and removal of riparian vegetation and large woody debris, resulted in dramatic alterations in the geography of the construction sites [11]. These changes have led to physico-chemical and habitat alterations, as well as a shift from lotic to more lentic conditions in the upper part of the weirs. These anthropogenic impacts may directly affect aquatic organism (i.e., fish, planktons, and macroinvertebrates), as well as the biodiversity and food-web structure [12,13,14,15]. Therefore, understanding the linkage among aquatic organisms is important to manage and control artificial environments, such as weirs.

Gut-content analysis is a fundamental step in the determination of food-web structures [16], and microscopic identification (MI) has conventionally been used for the analysis of gut contents. However, most gut content-analysis studies based on MI have the following disadvantages: (1) ambiguous prey specimen identification because of extensive digestion, (2) the presence of unidentified partial tissues, (3) identification failure due to a lack of expert knowledge, and 4) low-level identification resolution (identification of levels only higher than the family or order level). In addition, MI is unsuitable for tiny predators, such as a Chironomidae larvae and rotifers [17,18]. Applying DNA sequence-based techniques to gut-content identification has recently increased identification resolution, particularly in marine ecosystems [19,20,21,22,23,24]. However, relatively few studies have used DNA barcoding for gut-content analysis in complex freshwater ecosystems [25,26,27] although this technique has been recognized as a promising tool for studying food-web interactions [28].

The objectives of this study were therefore to examine the pattern of prey selection (selectivity index [*E*’]) by the 4th instar larvae of a Chironomidae using an MiSeq™ NGS platform (Illumina, San Diego, CA, USA). Chironomidae is omnivorous and plays an important role in the food chain of river ecosystems in Korea, where it is dominant. We aimed to evaluate (1) the pattern of food sources selection by Chironomidae in the large-scale weirs and (2) the applicability and effectiveness of the DNA meta-barcoding approach for identification of food selection by Chironomidae.

## 2. Materials and Methods

### 2.1. Study Area and Field Sampling

A survey was carried out at four study sites in the Gum River (SJ: Sejong Weir), Yeongsan River (JS: Juksan Weir), and Nakdong River (GG: Gangjeong-Goryeong Weir, DS: Dalseong Weir) from June to July 2019 (Figure 1; Appendix A, Table A1). We sampled the surface water for water quality (approximately the top 50 cm). Water temperature (Temp., °C), conductivity (Cond., µS/cm), dissolved oxygen (DO, mg/l), pH, and turbidity (NTU) were measured on-site using portable equipment (Model: YSI Professional Plus, Ohio, USA). The nutrients, chlorophyll-*a* (Chl-*a*), and carbon concentrations were analyzed in the laboratory. For total phosphorus (TP), total nitrogen (TN), and Chl-*a* concentration measurements, water samples were first filtered through a 0.45 μm pore-size membrane (Model: Advantec MFS membrane filter, Dublin, USA) and measurements were then performed using a UV spectrophotometer. Dissolved organic carbon (DOC) and total organic carbon (TOC) concentrations were measured using a TOC analyzer (Model: vario TOC cub, Langenselbold, Germany) through an 850 °C combustion catalytic-oxidation method. To collect 4th instar larvae from Chronomidae individuals, we used a Surber net (25 cm × 20 cm), dredging (1 m × 1 m), Ekman grab, and Ponar grab. After capture, the 4th instar larvae Chronomidae individuals were preserved in 96% ethanol and stored at room temperature for laboratory DNA meta-barcoding analysis.

### 2.2. DNA Extraction of Gut Contents and Metagenomic Sequencing

The gut contents were removed from the guts of 4th instar larvae of a Chironomidae (n = 12, 3 individuals each study site), and the dissection process was carried out after the complete volatilization of ethanol following the steps listed in Appendix A, Table A2. Genomic DNA was extracted using a DNeasy Blood & Tissue Kit (Cat. No. 69504, Qiagen, Düsseldorf, Germany) according to the manufacturer’s protocol. gDNA extracted for sequencing was prepared according to Illumina 18S Metagenomic Sequencing Library protocols (San Diego, USA). DNA quantity, quality, and integrity were measured using PicoGreen (Thermo Fisher Scientific, Waltham, USA) and a VICTOR Nivo Multimode Microplate Reader (PerkinElmer, Ohio, USA).

We selected two regions to amplify in the gDNA extracted from 4th instar larvae of a Chironomidae gut contents: the V9 region of the 18S rDNA gene (18S V9), primarily because of its broad range among eukaryotes [29,30]. The 18S rRNA gene was amplified using primers including an adaptor sequence: Forward Primer: 5’ TCGTCGGCAGCGTCAGATGTGTATAAGAGACAGC- CCTGCCHTTTGTACACAC 3’ / Reverse Primer: 5’ GTCTCGTGGGCTCGGAGATGTGTATAAGA- GACAGCCTTCYGCAGGTTCACCTAC 3’. First, to amplify the target region corresponding to the adapters, one cycle of 3 min at 95 °C; 25 cycles of 30 s at 95 °C, 30 s at 55 °C, and 30 s at 72 °C; and a final step of 5 min at 72 °C were carried out using the 18S V9 primers. Second, to perform indexing PCR, the first PCR product was amplified using one cycle of 3 min at 95 °C; 8 cycles of 30 s at 95 °C, 30 s at 55 °C, and 30 s at 72 °C; and a final step of 5 min at 72°C. The final products were normalized and pooled using PicoGreen (Thermo Fisher Scientific, USA), and the sizes of the libraries were verified using the LabChip GX HT DNA High Sensitivity Kit (PerkinElmer, Massachusetts, USA).

The library was sequenced using the MiSeq™ NGS platform (Illumina, San Diego, CA, USA) provided as a commercial service (Macrogen Inc., Seoul, Korea). Raw reads were trimmed using CD-HIT-OTU [31], and chimeras were identified and removed using rDnaTools. For paired-end merging, FLASH (Fast Length Adjustment of Short reads) version 1.2.11 was used [32]. Merged reads were processed using Qiime version 1.9 [33] and were clustered into operational taxonomic units (OTUs) using UCLUST [34] with a greedy algorithm employing OTUs at a 97% OTU cutoff value. Taxonomic classifications were assigned to the obtained representative sequences using BLASTn [35] and UCLUST [34].

### 2.3. Data Collection and Statistical Analysis

We implemented a literature survey to determine potential prey phytoplankton of Chironomidae larvae at the study sites [34]. To determine the prey selectivity of Chironomidae larvae, we calculated the selectivity index (*E*’) [36] using the relative abundances in water (from where Chironomidae larvae was obtained) of phytoplankton species in the gut content of Chironomidae larvae. An *E*’ over 0.5 indicated a preference for a prey.

## 3. Results

### 3.1. Meta-Barcoding and Taxonomic Assignment of Operational Taxonomic Unit (OTU)

In total, 1,019,526 paired-end reads were generated from the eight samples using 18SV9 primer sets on the Illumina MiSeq™ platform (Illumina, San Diego, CA, USA); of these, 98.0% passed Q30 (Phred quality score > 30) for improving the accuracy of sequences in this study. Each sample yielded 60,767–108,924 paired-end reads (mean: 89,970 reads), similar to the number of reads reported in a previous study [37], and all samples exhibited saturation of the number of OTUs by rarefaction curve analysis. Gamma-diversity was 381 OTUs, which were produced with a similarity cutoff of 97%. The resulting 381 OTUs were classified into 21 species- or genus-level taxonomic groups (those presenting < 0.1% abundance were removed). Uncultured and non-assigned reads were discarded.

After the assignment was performed, OTUs belonging to 16 orders, including 17 families, were found based on a BLASTn search of the NCBI database (Table 1). The abundances of the assigned sequences showed different patterns among the individuals (Figure 2). The dominant OTU was assigned to *Paractinolaimus* sp. (Nematoda), and the sub-dominant OTU was assigned to *Dicrotendipes fumidus* (Chironomidae). However, the most common OTUs among the individuals included phytoplankton, such as *Tetrahymena* sp., *D. armatus*, *Pseudopediastrum* sp., *Tetradesmus dimorphus*, *Biddulphia tridens*, and *Desmodesmus* sp. (Figure 2). Interestingly, we found an OTU sequence that was common among all individuals from *Hemibarbus labeo*, which is a benthic fish (Table 1).

### 3.2. Ecological Traits Based on Selectivity Index (E’) and Water Quality

Chironomidae larvae mainly consumed planktonic prey. Calculating the selectivity index (*E*’) using the relative abundance of gut-content phytoplankton species in the water from which Chironomidae larvae were obtained revealed differences in prey selectivity (Table 2) [36]. Chironomidae larvae showed a significant preference (*E*’ > 0.5) for *D. armatus*, *E. minima*, and *T. dimorphus*, while it showed negative selection for other species, even the ones that were found in its gut.

*N. palea* and *E. minima* appeared with higher DO, pH, and Chl-*a* values and *Tetrahymena* sp. appeared with higher NTU and TP values and lower Cond., TN, DOC, and TOC values, from the other prey sources OTUs (Table 1 and Table 3). Nutrient-related factors (Chl-a, TN, and TP) were related with *N. palea* and *E. minima* (small single cells), and carbon-related factors (DOC and TOC) were negatively related with *Tetrahymena sp*. (free swimming but chemokinetic cells).

## 4. Discussion

### 4.1. Prey Preference of Chironomidae Larvae

Our study provides scientific evidence that Chironomidae larvae have a significant preference (*E*’ > 0.5) for *D. armatus*, *E. minima*, and *T. dimorphus*, while showing negative selection for other species, even ones that are found in its gut (Table 2). These results coincide closely with those obtained in a tropic area by Henriques-Oliveira et al. [38] and Galizzi et al. [39], who reported that the Chironomidae family mainly consumes Bacillariophyta and Chlorophyta in the Tiradero River and Paraná River. We also found different patterns of OTUs’ relative abundance among the study sites (Table 1, Figure 2). Chironomidae showed a relatively high abundance for *D. armatus* in all study sites. We therefore suggest that Chironomidae prefers phytoplankton and that this preference is not affected by climate or natural environmental factors. However, differences in physio-chemical factors, such as water quality, nutrients, Chl-*a*, and carbon concentrations, caused by anthropogenic impacts (i.e., construction of large-scale weirs), particle size of prey species (small-sized single cell), and chemical factors (chemokinesis) can affect the feeding behavior of the 4th instar larvae of a Chironomidae. The diet preference of aquatic organisms hinders us in the application of environmental monitoring. Our results imply that application of NGS can be an alternative method to identify diet preferences of the macroinvertebrate organisms.

### 4.2. Efficiency of Meta-Barcoding for Analysis of Chironomidae Larvae Gut Contents

The DNA meta-barcoding approach for analyzing the gut contents of Chironomidae larvae have two advantages over other methods: (1) the size range of Chironomidae stages open to study has increased from adult to instar larvae and (2) prey identification to the species or genus level has become possible. Previously, small Chironomidae species stages, including instar larvae, could not be studied because their guts were too small to be examined by MI. Samples obtained from this larval stage of Chironomidae had to be analyzed using methods requiring great expertise. Therefore, despite the importance of assessing prey in terms of predator size [40], MI often ignored the larval stage of Chironomidae. However, if surgical evisceration of the gut is possible for both larvae and adults, their gut content analysis can be successfully carried out (Table 1 and Table 2); thus, a detailed understanding of the effects of Chironomidae populations on the biodiversity in an ecosystem can be based on analysis accounting for the size of species at all stages of growth [41]. The second advantage of DNA meta-barcoding is the high resolution of prey identification. MI is often impeded by the incomplete nature of the prey specimen, and digestion degrades prey specimens, resulting in identification failure. These problems can be overcome by DNA meta-barcoding. The high-resolution characterization of the food web structure is possible, and detailed remedial strategies for species management (of both predator and prey) can be achieved. Of course, the impact of predators on prey species should also be investigated quantitatively in conjunction with the qualitative identification of prey species using DNA meta-barcoding.

### 4.3. Potential Indicator of the Surrounding Environment

Interestingly, we found OTUs from *H. labeo*, which is a benthic fish, in Chironomidae individuals among all weirs (Table 1), even though Chironomidae cannot consume *H. labeo* directly. Chironomid species are known to eat castoff cells from benthic fish as a source of food. Some studies have shown that MI of the gut contents of aquatic organisms can be used to supplement the biodiversity inventories of benthic macroinvertebrates [42,43,44,45]. Few studies have examined the potential of using aquatic organisms in gut contents to monitor ecosystem function using the DNA meta-barcoding approach [46]. However, none of these studies have examined the effectiveness of the procedure for monitoring or evaluating biodiversity. DNA meta-barcoding not only provides dietary insights for estimating the impact of the food-web structure in large-scale weirs but also a tool to assess biological indicators. Additional experimental studies, based on the approach used in the present study, are necessary to develop a reliable biological indicator.

## Figures and Tables

**Figure 1 ijerph-17-02856-f001:**
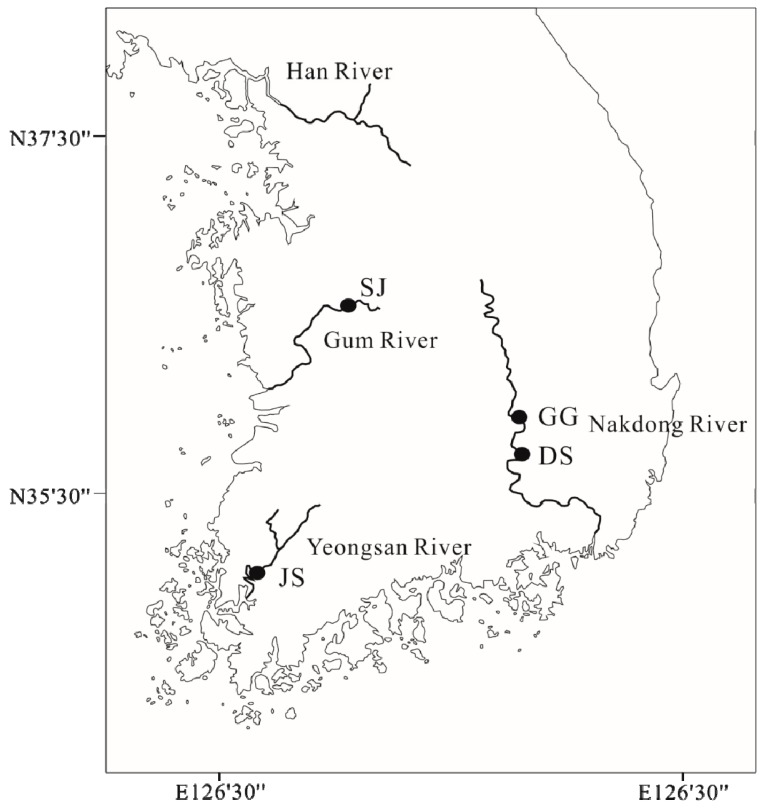
Map of the study sites in the Gum River (SJ: Sejong Weir), Yeongsan River (JS: Juksan Weir), and Nakdong River (GG: Gangjeong-Goryeong Weir, DS: Dalseong Weir).

**Figure 2 ijerph-17-02856-f002:**
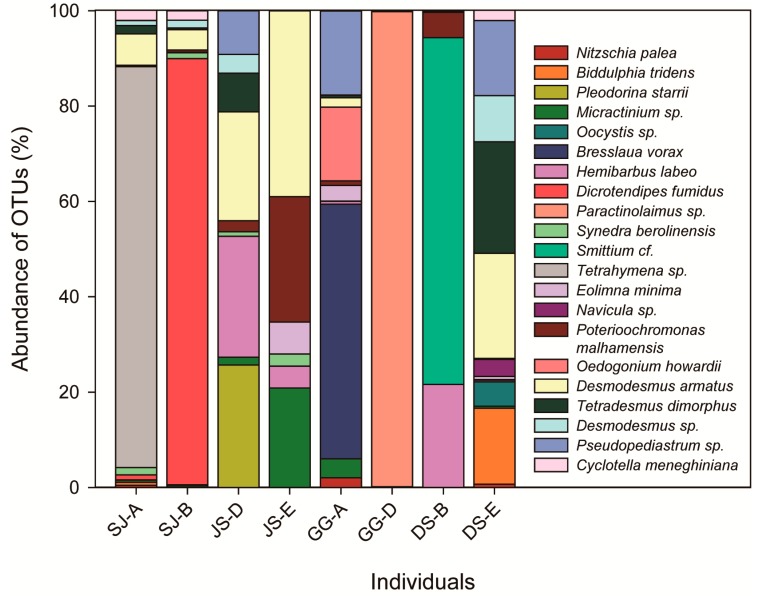
Abundance of OTUs among the individuals in the gut contents of Chironomidae based on the species or genus identification level (18SV9 regions, SJ.A-B: Sejong Weir, JS.D-E: Juksan Weir, GG.A-D: Gangjeong-Goryeong Weir, and DS.B-E: Dalseong Weir).

**Table 1 ijerph-17-02856-t001:** List of the Operational Taxonomic Unit (OTU) in the gut contents of Chironomidae based on the 18SV9 region (SJ.A-B: Sejong Weir, JS.D-E: Juksan Weir, GG.A-D: Gangjeong-Goryeong Weir, and DS.B-E: Dalseong Weir).

Order	Family	Genus + Species	SJ.A.	SJ.B.	JS.D.	JS.E.	GG.A.	GG.D.	DS.B.	DS.E.	Total	%	Identity	Query	Access ID
Ochromonadales	Ochromonadaceae	*Poterioochromonas malhamensis*	11	415	197	1,240	303	48	207	78	2499	0.6	99	100	MH536661.1
Fragilariales	Fragilariaceae	*Synedra berolinensis*	577	1088	81	120	5	4		5	1880	0.5	99	100	EF491890.1
Chlamydomonadales	Volvocaceae	*Pleodorina starrii*			2196	1			1		2198	0.5	99	99	LC086359.1
Chlorellales	Chlorellaceae	*Micractinium sp.*	174	318	140	983	1256	1		129	3,001	0.7	99	100	MF959935.1
	Oocystaceae	*Oocystis sp.*	4							1652	1,656	0.4	98	97	LC472542.1
Thalassiosirales	Stephanodiscaceae	*Cyclotella meneghiniana*	755	1720	1		22		4	671	3,173	0.8	99	100	AB430591.1
Naviculales	Naviculaceae	*Eolimna minima*	96	63		316	1026			210	1,711	0.4	99	100	AJ243063.2
		*Navicula sp.*	1							1146	1,147	0.3	99	100	FN398345.1
Oedogoniales		*Oedogonium howardii*					4885				4,885	1.2	99	100	EF616486.1
Sphaeropleales	Scenedesmaceae	*Desmodesmus armatus*	2416	3675	1951	1,837	623	63	4	7110	17,679	4.4	99	100	MK541798.1
		*Tetradesmus dimorphus*	644	290	690	1	165	38	2	7539	9,369	2.3	99	100	MN238814.1
		*Desmodesmus sp.*	386	1417	339		8	26	3	3132	5,311	1.3	99	100	MF326555.1
	Hydrodictyaceae	*Pseudopediastrum sp.*	6	2	781	1	5555		1	5069	11,415	2.8	99	100	KT883909.1
Biddulphiales	Biddulphiaceae	*Biddulphia tridens*	226	2			1	6		5152	5,387	1.3	96	100	JX401228.1
Bacillariales	Bacillariaceae	*Nitzschia palea*	171	72			646			219	1,108	0.3	98	100	KU948218.1
Colpodida	Colpodidae	*Bresslaua vorax*	1	2			16,835			1	16,839	4.2	98	100	AF060453.1
Hymenostomatida	Tetrahymenidae	*Tetrahymena sp.*	30,932	29	1		6			21	30,989	7.7	96	97	KX759198.1
Diptera	Chironomidae	*Dicrotendipes fumidus*	388	77,195		1	11			19	77,614	19.2	98	95	AY821866.1
Cypriniformes	Cyprinidae	*Hemibarbus labeo*	6	114	2,165	216	186	109	846	108	3,750	0.9	98	100	MH843153.1
Dorylaimida	Actinolaimidae	*Paractinolaimus sp.*						79,807			79,807	19.7	96	99	KM067902.1
Harpellales	Legeriomycetaceae	*Smittium cf.*							2845		2,845	0.7	96	99	JQ302895.1
		Number of sequences	36,794	86,402	8542	4716	31,533	80,102	3913	32,261	284,263				
		Number of OTUs	17	15	11	10	16	9	9	17	21				

**Table 2 ijerph-17-02856-t002:** Relative abundances and selectivity indexes (E’) of phytoplankton species present in the gut content of Chironomidae in the water from where Chironomidae was obtained.

Taxa	Relative Abundance of Prey		Selectivity Index (E’)
	Gut Contents (Ri)	In the Water (Pi)	
*Cyclotella meneghiniana*	8.85	73.53	−0.79
*Desmodesmus armatus*	49.32	3.98	0.85
*Eolimna minima*	4.77	0.53	0.80
*Navicula* sp.	3.20	1.38	0.40
*Nitzschia palea*	3.09	10.61	−0.55
*Oocystis* sp.	4.62	7.16	−0.22
*Tetradesmus dimorphus*	26.14	2.81	0.81

**Table 3 ijerph-17-02856-t003:** Detailed information of water quality along sites in the Gum River (SJ: Sejong Weir), Yeongsan River (JS: Juksan Weir), and Nakdong River (GG: Gangjeong-Goryeong Weir, DS: Dalseong Weir) (Water temperature (Temp., °C), conductivity (Cond., µS/cm), dissolved oxygen (DO, mg/l), pH, turbidity (NTU), phosphorus (TP), total nitrogen (TN), Dissolved organic carbon (DOC), total organic carbon (TOC) and chlorophyll-*a* (Chl-*a*)).

Study Sites	Temp.(°C)	Cond.(µS/cm)	DO (mg/L)	pH	NTU	TN	TP	TOC	DOC	Chl-*a*
SJ-A	24.6	321	5.01	8.27	23.70	2.813	0.081	3.2	2.1	31.0
SJ-B	20.7	287	4.50	7.60	23.00	2.434	0.142	3.8	2	5.2
JS-D	25.7	249	4.23	7.93	23.80	6.984	0.116	4.7	4.5	11.4
JS-E	26.0	246	4.25	8.03	25.12	4.106	0.106	5.0	4.5	23.0
GG-A	18.6	380	6.57	8.74	17.40	4.589	0.068	2.8	2.7	35.3
GG-D	27.4	496	3.93	8.66	3.49	2.150	0.045	3.5	3	2.9
DS-B	23.3	427	5.51	8.45	5.24	2.129	0.052	3.0	3	4.9
DS-E	24.4	402	8.04	9.34	9.18	3.574	0.067	3.0	3	11.0

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
