# Peer review of "First Gut Content Analysis of 4th Instar Midge Larvae (Diptera: Chronomidae) In Large-Scale Weirs Using a DNA Meta-Barcoding Approach"

_ijerph, 2020, doi:10.3390/ijerph17082856_

Round 1
Reviewer 1 Report
ijerph-760562
Overview
I recommend that the manuscript is not published at this time. Although the research has merit and is important, the manuscript needs to be revised in a few important areas prior to publication. The manuscript is well written and succinct, but the authors need to address major concerns about how they treat the study taxon, Chironomidae, as described in my comments below. The use of the meta-barcoding approach has promise and I hope that the manuscript will be published when the authors address the suggested revisions.
Use of Chironomidae as a study taxon. Chironomid larvae exhibit many different feeding behaviors. The authors establish that this family of aquatic Diptera have between 8000 and 20,000 species. On-line resources will show that food items range from small prey to detritus amongst this family. Thus, summarizing the gut contents of these 12 specimens at the family level, as a single “Chironomidae larva” should be avoided. Even identification at higher taxonomic levels such as subfamily and tribe would have provided some information if identification to genus and species was not possible. Given that the authors may not be able to make these identifications now, I suggest that they refer to their study organisms as “Chironomidae larvae”, in the plural, and make reference to the fact that the study organisms represent different taxa of Chironomidae. Thus, the gut content analysis represents food from a variety species, not just one Chironomdiae larva. Feeding strategies include grazing of algae, collecting and gathering of detritus, and predation, to name a few. This may explain why the results of the meta-barcoding analysis revealed benthic fish material in the guts of the study organisms. It is possible that some of the chironomid taxa, possibly detritivores, were consuming cells cast off from these fish.
Environmental analysis. Any environmental analysis will be misleading since you do not identify Chironomidae below the family level (see comments below). Perhaps you could keep the chemical data but show it in a table instead as a CCA with the gut contents. I suggest that this paper serve as a methods paper only. It is important enough to stand on its own merit as a methods paper.
Specific comments
Title
Please change “First gut content analysis of 4th instar larva of a Chronomids in large-scale weirs using a DNA meta-barcoding approach” to “First gut content analysis of 4th instar midge larvae (Diptera: Chironomidae) in large-scale weirs using a DNA meta-barcoding approach”
Abstract
Line 12. Chironomidae larva indicates a single taxon or individual organism. Please change this to Chironomidae larvae.
Introduction
Line 36. I suggest using another source/citation for this information other than [9]. Citation [1] provides many details on the life history of Chironomdiae.
Lines 48-50. Please change “These anthropogenic impacts may not only directly affect
aquatic organism (i.e. fish, planktons, and macroinvertebrates), but also the biodiversity and
food-web structure [12,13,14,15].” to “These anthropogenic impacts may directly affect
aquatic organism (i.e. fish, planktons, and macroinvertebrates), as well as the biodiversity and
food-web structure [12,13,14,15].”.
Line 57. Please include one or more citations to support the list of disadvantages related to VI of gut contents. These seem to be true for fish gut content analyses, but I have not seen research about gut contents on Chironomidae that support these claims. Most researchers who study chironomids slide-mount at least some of the specimens to make genus and species level identifications. Once slide mounted, algae and macroinvertebrate prey may be easily identified. Perhaps you may substitute that VI is difficult in such small insects despite their overall importance in food webs (with citations).
Materials and Methods
Lines 90 and 119. Given that you only have 12 specimens, I suggest that this paper serve only as a methods paper. You may have difficulty establishing environmental relationships with such a small “N”. Given that you did not identify the Chironomidae to subfamily, tribe, genus or species, it is possible that the pattern you observe in the CCA is related to specific taxa rather than environmental variables.
Author Response
Reviewer 1.
Overview
I recommend that the manuscript is not published at this time. Although the research has merit and is important, the manuscript needs to be revised in a few important areas prior to publication. The manuscript is well written and succinct, but the authors need to address major concerns about how they treat the study taxon, Chironomidae, as described in my comments below. The use of the meta-barcoding approach has promise and I hope that the manuscript will be published when the authors address the suggested revisions.
- First of all, we appreciate your comments for our manuscript. We tried to do our best to revise our paper according to your comments. Please see the revised manuscript. Changes are marked in RED.
Use of Chironomidae as a study taxon. Chironomid larvae exhibit many different feeding behaviors. The authors establish that this family of aquatic Diptera have between 8000 and 20,000 species. On-line resources will show that food items range from small prey to detritus amongst this family. Thus, summarizing the gut contents of these 12 specimens at the family level, as a single “Chironomidae larva” should be avoided. Even identification at higher taxonomic levels such as subfamily and tribe would have provided some information if identification to genus and species was not possible. Given that the authors may not be able to make these identifications now, I suggest that they refer to their study organisms as “Chironomidae larvae”, in the plural, and make reference to the fact that the study organisms represent different taxa of Chironomidae. Thus, the gut content analysis represents food from a variety species, not just one Chironomdiae larva.
- We agree with your precise comments. We changed the word from “larva” to “larvae” throughout the manuscript.
Feeding strategies include grazing of algae, collecting and gathering of detritus, and predation, to name a few. This may explain why the results of the meta-barcoding analysis revealed benthic fish material in the guts of the study organisms. It is possible that some of the chironomid taxa, possibly detritivores, were consuming cells cast off from these fish.
- Yes, we agree with your opinion and added some sentence at line 206 to explain it. We also have a plan to carry out additional experiments for further study which is focused on your aspect.
Environmental analysis. Any environmental analysis will be misleading since you do not identify Chironomidae below the family level (see comments below). Perhaps you could keep the chemical data but show it in a table instead as a CCA with the gut contents. I suggest that this paper serve as a methods paper only. It is important enough to stand on its own merit as a methods paper.
- We accepted your opinion. We excluded environmental analysis such as CCA and provided the chemical data in Table 3.
Specific comments
Title
Please change “First gut content analysis of 4th instar larva of a Chronomids in large-scale weirs using a DNA meta-barcoding approach” to “First gut content analysis of 4th instar midge larvae (Diptera: Chironomidae) in large-scale weirs using a DNA meta-barcoding approach”
- We revised the title according to your comment. Please see the line 2-3.
Abstract
Line 12. Chironomidae larva indicates a single taxon or individual organism. Please change this to Chironomidae larvae.
- We revised the word at line 12, but also throughout the manuscript.
Introduction
Line 36. I suggest using another source/citation for this information other than [9]. Citation [1] provides many details on the life history of Chironomdiae.
- We revised citation information.
Lines 48-50. Please change “These anthropogenic impacts may not only directly affect aquatic organism (i.e. fish, planktons, and macroinvertebrates), but also the biodiversity and food-web structure [12,13,14,15].” to “These anthropogenic impacts may directly affect aquatic organism (i.e. fish, planktons, and macroinvertebrates), as well as the biodiversity and food-web structure [12,13,14,15].”.
- We revised the sentence according to your comment (line 48-50).
Line 57. Please include one or more citations to support the list of disadvantages related to VI of gut contents. These seem to be true for fish gut content analyses, but I have not seen research about gut contents on Chironomidae that support these claims. Most researchers who study chironomids slide-mount at least some of the specimens to make genus and species level identifications. Once slide mounted, algae and macroinvertebrate prey may be easily identified. Perhaps you may substitute that VI is difficult in such small insects despite their overall importance in food webs (with citations).
- We included new references [17,18] to support the list of disadvantages related to VI of gut contents. Our paper is the first gut content analysis of midge larvae using a DNA meta-barcoding approach which have high identification level and broad range of targeted organisms.
Materials and Methods
Lines 90 and 119. Given that you only have 12 specimens, I suggest that this paper serve only as a methods paper. You may have difficulty establishing environmental relationships with such a small “N”. Given that you did not identify the Chironomidae to subfamily, tribe, genus or species, it is possible that the pattern you observe in the CCA is related to specific taxa rather than environmental variables.
- We agreed with your comment and revised our paper as a method paper only. We have a plan to carry out new approaches to analyze gut contents of midge larva based on the species level identification using DNA barcoding approach.
Reviewer 2 Report
This study investigated a diet of Chironomidae larva inhabiting four large-scale weirs in the Republic of Korea. It was shown that Chronomidae larva has a significant preference to some species and that its diet is also affected by physio-chemical factors and size of prey organisms. It was argued that VI visual inspection of gut contents suffers from significant drawbacks. Current study applied DNA barcoding for the gut-content analysis in freshwater ecosystem as a tool to investigate food-web interactions and showed its great potential in examining gut contents and elucidating patterns of Chironomidae prey selection in aquatic ecosystems. It is interesting ad useful study. This work would improve even more if in the discussion the authors would provide a perspective on a PRACTICAL value and significance of their findings with respect to environment and ecology. Below is a summary of the points that should be addressed. 1. Abstract line 21 (E`>0.5) few sentence explanation of E' . 2. Line 133 please provide more details about 8 samples. How samples were taken same or different time? 3. Line 149 Table 1: There are quite large differences between the numbers of sequences from different regions. For example DS.B region has 3,913 sequences, but the same region DS.E has 32,261 sequences. Why such a big difference? Also JS.D and JS.E have much less seqquences than other regions. Is there an explanation of these differences? 4. Line 152 Figure 2. The OTUS from the samples of the same sites are very heterogeneous. Would not we expect that the same sites are more similar to each other in OTUs composition, while the different geographic sites differ in the OTUs composition? Is there an explanation for the high heterogeneity i OTUs composition from the same geographic site? May the sampling method impacts seeing such results? In Figure 2 the label on x-axis Individuals - is this label correct? 5. Starting Line 155, Section 3.2. Canonical Correspondence Analysis. Which data was used in CCA? Is it the count data of Table 1? If not, please provide details and if possible a spreadsheet with counts as a Supplementary data. This is important for the reproducibility of the results. 6. Line 174. Please provide a reference to the method of calculation of the selectivity index. 7. Lines 208-209 The sentence " The diet preference ..." is confusing, please simplify or make a clearer statement. 8. Line 243 indicates Supplementary items that that are not referenced in the main text (for example Video). Please correct.Author Response
Reviewer 2.
This study investigated a diet of Chironomidae larva inhabiting four large-scale weirs in the Republic of Korea. It was shown that Chronomidae larva has a significant preference to some species and that its diet is also affected by physio-chemical factors and size of prey organisms. It was argued that VI visual inspection of gut contents suffers from significant drawbacks. Current study applied DNA barcoding for the gut-content analysis in freshwater ecosystem as a tool to investigate food-web interactions and showed its great potential in examining gut contents and elucidating patterns of Chironomidae prey selection in aquatic ecosystems. It is interesting ad useful study. This work would improve even more if in the discussion the authors would provide a perspective on a PRACTICAL value and significance of their findings with respect to environment and ecology.
- We appreciate your positive comments for our paper. We tried to do our best to revise our paper according to your comments. Please see the revised manuscript. Changes are marked in RED.
Below is a summary of the points that should be addressed.
- Abstract line 21 (E`>0.5) few sentence explanation of E' .
- We added some sentence to explain E’ (line 20-21).
- Line 133 please provide more details about 8 samples. How samples were taken same or different time?
- We carried out sampling at four study sites from June to July 2019 (line 72-74). Please see the detailed information of samples in Table S.2
- Line 149 Table 1: There are quite large differences between the numbers of sequences from different regions. For example DS.B region has 3,913 sequences, but the same region DS.E has 32,261 sequences. Why such a big difference? Also JS.D and JS.E have much less seqquences than other regions. Is there an explanation of these differences?
- Each sample yielded 60,767–108,924 paired-end reads (mean: 89,970 reads) by the MiSeqTM NGS platform, similar to the number of reads reported in a previous study [36]. We removed those presenting < 0.1% abundance of sequences, but also uncultured and non-assigned reads were discarded. These procedures caused large variation of sequence number between the samples.
- Line 152 Figure 2. The OTUS from the samples of the same sites are very heterogeneous. Would not we expect that the same sites are more similar to each other in OTUs composition, while the different geographic sites differ in the OTUs composition? Is there an explanation for the high heterogeneity i OTUs composition from the same geographic site? May the sampling method impacts seeing such results?
- We used the Chironomidae larvae samples. Chironomidae larvae exhibit many different feeding behaviors and show that food items range from small prey to detritus amongst this family. We think these samples are not same species. Therefore, we have a plan to carry out additional experiments for further study to determine the prey preference following Chironomidae species in various sites.
In Figure 2 the label on x-axis Individuals - is this label correct?
- Yes it is right.
- Starting Line 155, Section 3.2. Canonical Correspondence Analysis. Which data was used in CCA? Is it the count data of Table 1? If not, please provide details and if possible a spreadsheet with counts as a Supplementary data. This is important for the reproducibility of the results.
- We excluded environmental analysis such as CCA according to reviewer 1’s opinion and provided table of the chemical data. Please see the Table 3.
- Line 174. Please provide a reference to the method of calculation of the selectivity index.
- We added a reference [35].
- Lines 208-209 The sentence " The diet preference ..." is confusing, please simplify or make a clearer statement.
- We simplified the sentences (line 181-182).
- Line 243 indicates Supplementary items that are not referenced in the main text (for example Video). Please correct.
- We corrected it.
Round 2
Reviewer 2 Report
The authors answered my comments.